# MicroRNAs in Hypertrophic, Arrhythmogenic and Dilated Cardiomyopathy

**DOI:** 10.3390/diagnostics11091720

**Published:** 2021-09-19

**Authors:** Enrica Chiti, Marco Di Paolo, Emanuela Turillazzi, Anna Rocchi

**Affiliations:** 1Institute of Life Science, Scuola Superiore Sant’Anna, 56124 Pisa, Italy; enricachiti@gmail.com; 2Department of Surgical Pathology, Medical, Molecular and Critical Area, Institute of Legal Medicine, University of Pisa, Via Roma 55, 56126 Pisa, Italy; marco.dipaolo@unipi.it (M.D.P.); emanuela.turillazzi@unipi.it (E.T.)

**Keywords:** microRNAs, biomarkers, inherited cardiomyopathies, dilated cardiomyopathy, arrhythmogenic cardiomyopathy, hypertrophic cardiomyopathy

## Abstract

MicroRNAs (miRNAs) are a class of non-coding RNAs of about 20 nucleotides in length, involved in the regulation of many biochemical pathways in the human body. The level of miRNAs in tissues and circulation can be deregulated because of altered pathophysiological mechanisms; thus, they can be employed as biomarkers for different pathological conditions, such as cardiac diseases. This review summarizes published findings of these molecular biomarkers in the three most common structural cardiomyopathies: human dilated, arrhythmogenic and hypertrophic cardiomyopathy.

## 1. Introduction

Cardiomyopathies are a very heterogeneous group of cardiac diseases, and the inherited forms have a prevalence of 1:250 to 1:5000 in the general population depending on the subtype [1], representing a frequent cause of cardiac arrest and sudden cardiac death (SCD), especially in the young [2].

Genetically inherited cardiomyopathies represent a significant percentage of cardiovascular disorders [3]. The main inherited cardiomyopathies are hypertrophic cardiomyopathy (HCM), arrhythmogenic cardiomyopathy (ACM), and dilated cardiomyopathy (DCM) [4].

In the last twenty years, important developments have been made in the understanding of the underlying pathogenetic mechanisms of these cardiac diseases.

Only a small percentage of patients with genetically inherited cardiomyopathies carry known genetic mutations, therefore, there is the possibility that other mechanisms are implicated in their development.

Several circulating and cardiac miRNAs have been found deregulated in genetic cardiomyopathies [5,6].

MiRNAs are biomolecules with a crucial role in the physiological development of the heart and are involved in cardiovascular diseases such as coronary artery diseases (CAD), leading SCD cause in adult subjects [7], heart failure and left ventricular hypertrophy [8], atrial fibrillation [9] and stroke [10].

MiRNAs are small non-coding RNAs (about 20 nt in length) that regulate the gene expression at post-transcriptional level. They bind complementary mRNA sequences silencing them by preventing protein synthesis or via degradation through mRNA cleavage. Imperfect base-pairing results in repression of mRNA translation while perfect base pairing determines mRNA cleavage. MiRNAs maturation starts in the nucleus with primary miRNAs (pri-miRNAs) which are processed into pre-miRNAs by Drosha and then exported into the cytoplasm by exportin 5. RNase III enzyme cleaved pre-miRNAs in duplex miRNA of 22 nucleotides in length: one strand is incorporated into the RNA-induced silencing complex (RISC) while the other strand is normally degraded. Mature miRNA incorporated in RISC is less susceptible to degradation processes [11].

The human genome encodes more than 1900 miRNAs, as reported by the biological database *MiRbase* (www.mirbase.org).

Most miRNAs are located within the cell, but some of them are present in various body fluids and are called circulating miRNAs [12].

Circulating miRNAs are promising biomarkers, highly stable at extreme conditions of pH, chemical treatments and temperature. Moreover, circulating miRNAs are preserved from RNAse activity because they are carried by RNA-binding proteins in extracellular vesicles and lipoprotein [13,14].

Impaired miRNA expression is implicated in various disease states including cancer [15], metabolism disorders [16], and several neurodegenerative disorders [17].

The role of these biomolecules has also been strongly investigated in heart biology. Several studies reported miRNAs involvement in cardiovascular diseases such as heart failure and left ventricular hypertrophy [8] or atrial fibrillation [9]. Their expression levels (in circulation and/or in cardiac tissue) can be found deregulated in pathological conditions [5].

This review summarizes the latest miRNAs studies conducted on human samples of the most common cardiomyopathies.

## 2. Article Selection

Literature search was performed on PUBMED and Google Scholar databases. “Human cardiomyopathy”, “Hypertrophic cardiomyopathy”, “HCM phenocopies” “Dilated cardiomyopathy”, “Arrhythmogenic cardiomyopathy” and “miRNAs” terms were combined to detect all the papers that investigated miRNAs expression in human samples of the three most frequent human structural cardiomyopathies. Finally, articles were selected after initial screening by title and abstract and then by full-text evaluation.

## 3. miRNAs as Potential Biomarkers in Hypertrophic Cardiomyopathy

Hypertrophic cardiomyopathy (HCM) is a structural cardiac disease characterized by left ventricular hypertrophy and a non-dilated ventricle in absence of other diseases able to determine the degree of hypertrophy. Cardiac myocytes are separated by interstitial fibrotic areas and lose their shape and their normal parallel alignment [18]. Disarray is a typical structural microscopic characteristic of HCM. In adult subjects, HCM is characterized by a wall thickness ≥ 15 mm in one or more LV myocardial segments, measured by different imaging technique types (i.e., echocardiography, computed tomography, or cardiac magnetic resonance). More limited hypertrophy (13–14 mm) in different LV sections can be diagnostic in first-degree relatives of a patient with HCM [19,20,21]. HCM is an autosomal dominant disease, characterized by high heterogeneity and variable expressivity. Up to 60% of HCM patients carry mutations in genes encoding sarcomeric proteins [22,23]. The remaining 40% of HCM patients are sporadic.

Epigenetic changes, such as those changes performed by miRNAs, could be involved in the pathogenesis of HCM. Different studies analyzed miRNAs’ role in HCM development and cardiac remodeling.

### 3.1. Circulating miRNAs in HCM

Roncarati et al., aiming to characterize the circulating miRNA profile in a group of 41 HCM subjects, analyzed 21 miRNAs directly involved in angiogenesis, fibrosis, apoptosis, hypertrophy, and smooth muscle cell biology. miR-214, miR-16, and miR-1 were downregulated while miR-27a, miR-199a-5p, miR-26a, miR-145, miR-133a, miR-143, miR-199a-3p, miR-126-3p, miR-29a, miR-155, miR-30a, and miR-21 were upregulated in patients compared to controls. Besides this the authors, evaluating the correlation between miRNAs and left ventricular (LV) parameters (fibrosis and hypertrophy), found that miR-27a, miR-29a, and miR-199a-5p correlate with cardiac hypertrophy. Moreover, high levels of miR-29a significantly correlate with both cardiac fibrosis and hypertrophy [24].

Fang et al. performed an array of 84 circulating miRNAs in eight HCM patients and selected 14 miRNAs significantly upregulated in patients with diffuse myocardial fibrosis compared to controls. These 14 miRNAs, together with miR-29a-3p and miR-133a-3p, which have a well-established role in cardiac fibrosis, were further validated in a group of 55 HCM patients. The authors found low levels of miR-373-3p and miR-96-5p and high levels of 14 miRNAs (miR-18a-5p, miR-146a-5p, miR-30d-5p, miR-17-5p, miR-200a-3p, miR-19b-3p, miR-21-5p, miR-193-5p, miR-10b-5p, miR-15a-5p, miR-192-5p, miR-296-5p, miR-29a-3p, and miR-133a-3p) in patients with HCM and diffuse myocardial fibrosis compared to patients with non-diffuse myocardial fibrosis [25].

Derda et al. analyzed eight circulating miRNAs selected from the literature (miR-1, miR-21, miR-29a, miR-29b, miR-29c, miR-133a, miR-155, and miR-499) in four different groups of subjects (23 patients with non-obstructive HCM, 28 patients with obstructive HCM, 47 subjects with cardiac hypertrophy due to aortic stenosis, and 22 controls). miR-155 was downregulated in both HCM subgroups (but not in patients with aortic stenosis) while miR-29a was significantly upregulated only in obstructive HCM patients. High levels of circulating miR-29c exclusively characterized the aortic stenosis group. In the same study the researchers evaluated circulating miRNAs levels in obstructive HCM due to sarcomeric mutations in MHY7 and MYBPC3 genes: miR-29a levels were increased in obstructive HCM carrying MYH7 mutation but not in MYBPC3 patients [26].

Zhou et al. evaluated the expression levels of some potentially relevant miRNAs and long non-coding RNAs (lncRNAs) in serum samples of HCM patients with and without fibrosis and found high levels of miR-29a in the fibrosis HCM group compared to subjects without fibrosis. In contrast, patients without fibrosis showed high levels of the lncRNA MIAT. Moreover, wild-type MIAT cells, transfected with miR-29a mimics, showed a suppressed luciferase activity, suggesting a negative regulation of miR-29a by MIAT [27].

Gudkova et al. found instead that miR-21 levels were higher in HCM patients than in controls [28].

### 3.2. Cardiac miRNAs in HCM

miR-590–5p and miR-92a were higher while miR-1, miR-133b, miR-191, miR-208b, miR-218, miR-30b, miR-374, miR-454, and miR-495 were lower in the cardiac tissues of five HCM subjects (three with left ventricular hypertrophy caused by heart valve disease and two with MYH7 mutations) compared to controls [29].

Wang et al. reported higher levels of cardiac miR-221 in HCM compared to controls [30].

Leptidis et al. evaluated five heart-failure HCM patients and found twenty miRNAs (miR-1-3p, miR-23a-3p, miR-23b-3p, miR-24-3p, miR-29b-3p, miR-30d-5p, miR-125a-5p, miR-126-3p, miR-133a-3p, miR-143-3p, miR-145-5p, miR-193b-3p, miR-197-3p, miR-331-3p, miR-342-3p, miR-361-5p, miR-365-3p, miR-455-3p, miR-1975-3p, miR-1978) upregulated in patients compared to controls [31].

Contrasting results were obtained by Liu et al. about miR-29 family levels, with no differences between HCM patients and controls [32].

Kuster et al. described a specific miRNA HCM profile in six patients carrying MYBPC3 mutation. miR-10b and miR-10b* were downregulated while miR-184, miR-497, miR-204, miR-34b*, miR-222* were upregulated in HCM patients compared to controls. In silico mRNA target prediction demonstrated that many of the miRNA target genes were related to β-adrenergic pathway (a part of cardiac hypertrophic signaling) [33].

Song L. et al. found that miR-21, miR-130b, and miR-132 were significantly upregulated, whereas miR-451, miR-363, miR-150, miR-3141, miR-144, miR-144*, miR-139-5p, miR-139-3p, miR-1246, miR-486-3p were downregulated compared to controls [34]. miR-139-5p downregulation in human HCM cardiac tissue was also confirmed by Ming and colleagues [35].

Low levels of miR-27a and miR-1-3p characterized instead transplanted hearts of both HCM and DCM subjects. In particular, miR-1-3p is inversely correlated with LV end-diastolic diameter and directly with LV ejection fraction and [36].

Sun et al. evaluating 367 differentially expressed miRNAs in five HCM cardiac tissues found miR-20 as one of the highly expressed miRNAs in the HCM group compared to controls [37].

More recently, Huang et al. found lower levels of miR-19b and miR-155 and higher levels of miR-221, miR-222, and miR-433 in 42 obstructive HCM patients compared to controls. In particular, cardiac miR-221 positively correlates with myocardial fibrosis while miR-19b inversely correlates with myocardial fibrosis [38].

As shown in Table 1, only a few miRNAs overlap between studies on human HCM (i.e., miR-29, miR-21, miR-133, and miR-1).

In particular, miR-29 family is one of the most commonly reported miRNAs in HCM patients. High levels of circulating miR-29a distinguish HCM patients from controls [24,25,26,27] and miR-29b upregulation also characterizes the cardiac tissue of end-stage HCM patients [31].

High levels of miR-21 are reported in HCM blood [24,25,28] and cardiac tissue [31].

High levels of miR-133 are reported in blood [24,25] and myocardium [31]. Contrary results were obtained by Palacin et al. [29].

miR-1 downregulation characterized both blood [24] and HCM cardiac tissue [29,36]. In contrast, Leptidis showed miR-1 upregulation in cardiac tissue [31].

### 3.3. miRNAs and HCM Phenocopies

A phenocopy is a phenotype that can falsely mimic the disease. Different genetic disorders not determined by sarcomeric mutations have been associated with severe left ventricular hypertrophy and are called HCM phenocopies.

Another possible role of miRNAs could be the contribution in differential diagnosis between HCM and phenocopies.

Fabry disease (FD), an X-linked lysosomal storage disorder, is characterized by mutations in the GLA gene, encoding the *α*-galactosidase A enzyme, causing the absence or reduction of the activity of the lysosomal enzyme and a subsequent lysosomal accumulation of globotriaosilceramide (Gb3) and other glycosphingolipids in different cell types, such as cardiomyocytes. Fabry disease can involve the heart mimicking HCM.

Recently, Cammarata et al. found that some circulating miRNAs could be used to differentiate FD from controls (divided in healthy controls and subjects with left ventricular hypertrophy). Researchers found that miR-199a-5p and miR-126-3p levels were higher in FD than in controls with left ventricular hypertrophy. Lower miR-423-5p and miR-451a levels characterized instead FD compared to controls [39].

Low levels of circulating miR-26a-5p, -21-5p, -152-5p, 1307-5p characterized FD patients after enzyme replace therapy [40].

Cardiac amyloidosis, identified by the deposition and misfolding of insoluble proteins that affect the heart, possibly leads to severe heart failure and cardiac hypertrophy.

A recent work evaluated circulating miRNA levels in patients with transthyretin amyloidosis, with senile cardiac amyloidosis, with HF with reduced ejection fraction and in controls. miR-27a levels were significantly decreased in transthyretin amyloidosis compared to controls while miR-399-3p levels were higher in senile cardiac amyloidosis compared to the other three groups [41].

Pompe disease, another HCM phenocopy, is a metabolic disease linked to mutations in the GAA gene, with a deficiency of the GAA enzyme involved in the lysosomal glycogen degradation. High levels of circulating miR-133a characterized patients affected by Pompe disease [42].

More studies are needed to discover useful miRNAs to differentiate diseases mimicking HCM phenotype from HCM itself.

## 4. miRNAs as Potential Biomarkers in Arrhythmogenic Cardiomyopathy

Arrhythmogenic Cardiomyopathy (ACM) is a genetic cardiac disease characterized by progressive fibrofatty substitution of the right ventricular cardiomyocytes and ventricular wall thinning. Left ventricular involvement affects more than 50% of cases [43]. Histology usually presents cardiomyocyte death, inflammation, and fibro-adipose substitution [44]. ACM is characterized by the development of ventricular arrhythmias [45] and heart failure and can be a cause of sudden cardiac death, especially in the young [46,47,48].

ACM diagnosis is particularly challenging, because of incomplete penetrance and variable expressivity and can be performed using different techniques, such as electrocardiography (ECG), imaging techniques (echocardiography and magnetic resonance), and genetic evaluation [49].

The majority of genetic mutations causing ACM are located in desmosomal genes as desmoplakin (DP), plakoglobin (PG), plakophilin 2 (PKP2), and desmoglein 2 (DSG2) [50].

### 4.1. Circulating miRNAs in ACM

Sommariva et al., demonstrating low levels of circulating miR-320a in 36 ACM patients compared to idiopathic ventricular tachycardia (IVT) patients and controls, hypothesized that miR-320a could be a promising biomarker to distinguish between arrhythmogenic cardiomyopathy and IVT. Moreover, miR-320a expression seemed not to be affected by heart remodeling/changes induced by physical exertion, a recognized ACM risk factor [51].

Yamada et al. studied circulating miRNAs in 62 patients with ventricular arrhythmia (VA): 23 with idiopathic ventricular tachycardia, 28 with diagnosed ACM and 11 with borderline or possible ACM. Higher levels of miR-144-3p, miR-145-5p, miR-185-5p, and miR-494 were found in plasma samples of ACM subjects with ventricular arrhythmia compared to controls, IVT patients and subjects with suspected arrhythmogenic cardiomyopathy. High levels of miR-494 correlate with recurrent ventricular arrhythmia after ablation in ACM subjects [52].

Recently, Bueno Marinas validated six miRNAs (miR-122-5p, miR-133a-3p, miR-133b, miR-142-3p, miR-182-5p, and miR-183-5p) in blood samples (n = 90 patients divided in ACM unaffected family members with a pathogenic variant, myocarditis, Brugada syndrome, hypertrophic cardiomyopathy and dilated cardiomyopathy). miR-122-5p, miR-182-5p, and miR-183-5p were higher in ACM compared to controls and the other cardiomyopathies, while miR-133a-3p, miR-133b, and miR-142-3p were downregulated in ACM than in controls and other groups [53].

### 4.2. Cardiac miRNAs in ACM

Zhang et al., in 2016, analyzed for the first time the cardiac tissue of 24 ACM patients subjected to cardiac transplant and found 12 upregulated (miR-21-3p, miR-21-5p, miR-34a-5p, miR-212-3p, miR-216a, miR-584-3p, miR-1251, miR-3621, miR-3674, miR-3692-3p, miR-4286, miR-4301) and 12 downregulated miRNAs (miR-135b, miR-138-5p, miR-193b-3p, miR-302b-3p, miR-302c-3p, miR-338-3p, miR-451a, miR-491-3p, miR-575, miR-3529-5P, miR-4254, miR-4643). MiR-21-5p and miR-135b were the most significantly deregulated. Researchers also established that target genes of miR-21-5p and miR-135b were involved in the *Wnt* and *Hippo* signaling pathways (BMPR2 is related to adipogenesis and TGFBR2 is associated with fibrosis and extracellular matrix production), molecular mechanisms probably related to the ACM molecular pathophysiology, suggesting a possible role in fibrofatty substitution [54].

Rainer et al. discovered higher levels of miR-29b-3p in cardiac stromal cells involved in adipogenesis of the ACM heart compared to controls [55]. Table 2 reports miRNAs potentially involved in human ACM.

## 5. miRNAs as Potential Biomarkers in Dilated Cardiomyopathy

Dilated cardiomyopathy (DCM) is another structural cardiac disease associated with heart failure and high risk of sudden death [56]. DCM is characterized by left ventricular wall thinning, systolic dysfunction, and chamber dilatation but right ventricular involvement is also observed. Extracellular matrix fibrosis is an important pathological DCM modification that can contribute to heart failure [57]. Different issues, such as myocarditis, drugs, alcohol, and genetic inheritance, are involved in DCM [58]. Most of the pathogenic mutations have been found in sarcomeric and nuclear envelope genes [59,60].

Diagnosis is based primarily on the exclusion of ischemic heart disease or chronic abnormal loading conditions such as hypertension and valvular disorders [61]. Other criteria are represented by the presence of fractional shortening less than 25% (>2SD) and/or ejection fraction less than 45% (>2SD), and left ventricular end-diastolic diameter (LVEDD) greater than 117% (>2SD of the predicted value of 112% corrected for age and body surface area) [62]. An electrocardiogram is the initial cardiac screening test used for suggestive (non-confirmatory) diagnosis of DCM. Echocardiography is the first-line imaging test and cardiac magnetic resonance is used to confirm the underlying diagnosis of cardiomyopathy [63].

Endomyocardial biopsy is recommended with suspicion of myocarditis or inflammatory cardiomyopathy in selected patients.

Prognosis is poor in DCM subjects with left ventricular ejection fraction (LVEF) of less than 35%: LVEF less than 35% is related to a high risk of sudden death [64].

Other factors, such as mitral regurgitation, fibrosis, or chamber enlargement, can worsen the prognosis [60].

The following scientific works evaluated miRNAs’ function in human DCM.

### 5.1. Circulating miRNAs in DCM

Nair et al. found low levels of circulating miR-142-3p and high levels of miR-124-5p in DCM patients [65].

Low levels of the miR-548 family (in particular miR-548c) characterized DCM subjects with stable chronic heart failure [66].

Fan et al. identified instead high levels of miR-423-5p in heart failure DCM patients [67].

Yu et al. found that miR-185 levels were higher in DCM patients compared to controls. Furthermore, high levels of miR-185 were associated with favorable prognosis in DCM patients because this miRNA repress the B cells involved in myocardial fibrosis and myocyte injuries [68].

Wang et al. described high levels of miR-3135b, miR-3908, and miR-5571-5p in human DCM. In particular, miR-5571-5p upregulation was significantly associated with symptom severity and disease progression [69].

A study conducted on five circulating miRNAs and extracellular matrix (ECM) fibrosis demonstrated a correlation between miR-26 and miR-30 and collagen volume fraction, related to fibrosis [70].

In DCM acute heart failure patients, miR-92b-5p levels were higher than in controls [71].

Onrat et al. found high levels of miR-24-3p, miR-28-5p, miR-100-5p, miR-103-3p, miR-125b5p, miR-214-3p, let-7b-5p, and let-7c-5p in ischemic and idiopathic DCM [72].

Toro et al. analyzed DCM patients with pathogenic LMNA mutations (one of the most common variants associated with familial DCM) and found increased levels of let-7a-5p, miR-142-3p, miR-145-5p, and miR-454-3p compared to idiopathic DCM or LMNA controls [73]. The same group, two years later, proposed a six-miRNA panel (let-7a-5p, let-7g-5p, miR-16-2-3p, miR-210-3p, miR-215-5p, and miR-629-5p) to differentiate DCM subjects with pathogenic BAG3 or LMNA variants from wild-type variant carriers. A second group of five miRNAs (miR-19b-3p, miR-29a-3p, miR-130b-3p, miR-215-5p, miR-629-5p) was proposed instead to distinguish phenotypically negative variant carriers from healthy subjects [74].

Zaragoza et al. analyzed a Spanish family affected by familial DCM due to BAG3 (BAG cochaperone 3) gene mutations: circulating miR-154-5p, miR-182-5p, miR-1249-ep, miR-3191-3p, miR-6769b-3p, and miR-6855-5p were increased in BAG3+ mutant carriers compared to BAG3 wild-type. Moreover, miR-154-5p, miR-182-5p and miR-6885-5p, correlates with systolic and diastolic blood pressure, A wave, left atrium area and length in BAG3+ mutation carriers [75].

Dziewięcka et al., in 2020, evaluated both circulating and cardiac miRNAs in left ventricular reverse remodeling (LVRR) DCM and found that only cardiac miR-133a showed a significantly increased expression in LVRR compared to non-LVRR patients [76].

A recent study conducted by Calderon-Dominguez in 2021 presented a predictive three-miRNAs model (high levels of miR-130b-3p, miR-150-5p, and miR-210-3p) that, combined with clinical variables, could distinguish idiopathic DCM with severe reduced systolic ejection fraction from patients with moderately reduced ejection fraction [77].

### 5.2. Cardiac miRNAs in DCM

Satoh et al. described higher levels of miR-21 and lower levels of let-7i, miR-126, and miR-155 in cardiac tissues of DCM patients compared to controls and patients without left ventricular dysfunction. Researchers demonstrated a relationship between low levels of let-7i and a poor clinical outcome (heart failure and death) [78]. High levels of miR-21 were also confirmed in human DCM and myocarditis animal models, highlighting a possible role of miR-21 in myocardial fibrosis development in these two cardiac diseases [79].

Greco et al. analyzed miRNA expression profiles in left ventricular biopsies from type-2 diabetic and non-diabetic subjects affected by ischemic DCM. In particular, miR-34b, miR-34c, miR-199b, and miR-210 were higher in diabetic DCM compared to controls, while miR-223 and miR-650 were higher in diabetic compared to non-diabetic DCM but lower in non-diabetic DCM compared to controls. Moreover, miR-216a resulted in upregulation in both DCM groups compared to controls. In particular, miR-216a expression inversely correlated with left ventricular ejection fraction [80].

Besler et al. observed higher miR-133a levels in inflammatory DCM without fibrosis than in patients with fibrosis. High levels of miR-133a were associated with a reduction in necrosis and fibrosis in myocytes and left ventricular functional recovery in inflammatory DCM [81].

Naga Prasad et al. analyzed end-stage heart failure DCM and found lower levels of miR-378, miR-1, miR-7, and miR-29b and higher levels of miR-342, miR-214, miR-125b, miR-145, and miR-181b compared to controls [82].

Wang et al. studied miRNAs profile in different cardiac regions (left, right ventricle, apex, and septum) and found higher levels of miR-21 and miR-29 family and lower levels of miR-133 family in the left and right ventricle, and the apex of DCM patients compared to controls. No significant differences were found in the miRNA profile of the septum [83].

Zhou et al. found higher levels of miR-208b in severe heart failure DCM (subjected to assist device implantation) compared to subjects with heart failure due to ischemic heart disease or myocarditis [84].

Li et al., analyzing HCM and DCM samples, demonstrated higher levels of miR-155, miR-10b, and miR-23a in both patient groups compared to controls. Different expression patterns between DCM and HCM patients were found for miR-21(upregulated in DCM), miR-214 (downregulated in DCM), and miR-27a and miR-1-3p (downregulated in HCM) [36].

Both cardiac and circulating miR-29 and miR-26 were, respectively higher and lower in the DCM cohort compared to controls. Only cardiac miR-133a was instead downregulated in patients compared to controls [85].

Table 3 shows that few miRNAs are in common between different studies.

miR-21, a microRNA involved in the proliferation of fibroblasts and cardiac fibrosis, showed upregulation in cardiac tissue of DCM [36,78,79,83].

Contrasting results are observed for miR-29, reported as downregulated [82,83] and also upregulated [85] in the cardiac tissue of DCM patients. Additionally, miR-133 reports contrasting results in cardiac tissue: it has variously been found to be downregulated [83,85] and also upregulated in cardiac tissue [81].

### 5.3. DCM Is Common Cardiomyopathy in Childhood

DCM represents a relevant issue both in the adult and pediatric population but only few studies analyzed miRNAs in children affected by DCM (Table 4).

Miyamoto et al. conducted a study on 55 children (<18 years) and found an upregulation of circulating miR-155 and miR-636 and a downregulation of miR-639 and miR-646 in DCM patients (who were transplanted or died), compared to DCM subjects with recovered ventricular function, suggesting the four miRNAs as diagnostic and prognostic biomarkers [86].

Enes Coşkun et al. analyzed plasma of idiopathic DCM children (2–192 months) and found a significant upregulation of miR-454 and miR-518f in patients than in controls. In the DCM children group, 10 circulating miRNAs (miR-99b, miR-147, miR-155, miR-194, miR-205, miR-218 miR-302a, miR-544, miR-618, and miR-875-3p) were instead downregulated [87].

Jiao and colleagues found higher levels of let-7f-5p, let-7g-5p, miR-26a-5p, miR-27a-3p, miR-27b-3p, miR-126-3p, miR-142-5p, and miR-143-3p in serum of child DCM patients compared to controls. In particular, increased levels of miR-126-3p and let-7g associated with a decreased ejection fraction compared with no heart-failure subjects [88].

Woulfe et al. analyzed instead left ventricular tissue of pediatric idiopathic DCM, characterized by lower levels of miR-29 family compared to controls. Significant downregulation of miR-29 was related to pediatric heart failure subjects without fibrosis [89].

Due to the difficulty of tissue sampling in children, circulating miRNAs could be promising biomarkers for DCM in young patients, together with their possible use as biomarkers for risk stratification in prognosis related to young people affected by DCM.

## 6. Discussion

miRNAs are biomolecules involved in many cardiovascular diseases, comprised of structural cardiomyopathies potentially involved in sudden cardiac death events. Identifying miRNAs that can be used as reliable biomarkers for the prevention and treatment of structural cardiomyopathies is an interesting challenge.

With recent advances in precision medicine, miRNAs that are differentially expressed in cardiac diseases could be considered potential markers

However, circulating miRNAs are more desirable to use as biomarkers in routine diagnosis because they are easily accessible compared to cardiac tissue miRNAs.

In fact, the complexity of sampling cardiac tissue represents a relevant limit. A possible solution of miRNA tissue analysis could be its evaluation in other biological samples. A promising candidate is represented by urine, as suggested by Cheng et al. in a study conducted on rat model of acute myocardial infarction (AMI). The authors found increased levels of heart-released miR-1 and miR-208 in urine samples after AMI [90]. High levels of cardiac-specific miR-1 were also confirmed by Duan et al. in the urine of ST elevation acute myocardial infarction (STEMI) patients compared to controls [91].

Furthermore, heart tissue miRNAs could be targets of cardiovascular drugs.

Studies conducted in animal models show that pharmacological treatments modulate miRNAs expression.

Zhao et al. analyzed the heart tissues of a cardiac hypertrophy mouse model and demonstrated that choline administration inhibited pathological cardiac hypertrophy, correcting miR-133a downregulation [92].

Tu et al. found instead that atorvastatin inhibited cardiomyocyte hypertrophy in rats, downregulating miR-22 and consequently modulating the activity of PTEN [93].

Additional research is needed to identify and validate miRNAs as diagnostic biomarkers.

Few studies have been conducted on miRNAs and humans affected by these cardiac diseases, and what emerges is that common results between studies are rare. The differences between studies could be due to different issues, such as a different number of samples/patients analyzed, the sampling time, miRNA quantification, and data normalization. The use of common operating procedures could reduce interstudy variability.

However, three miRNAs (miR-21, miR-29, and miR-133) are frequently reported as being deregulated in the analyzed studies, particularly in reports concerning human HCM and DCM. miR-21, miR-29, and miR-133 seem to be involved in the regulation of cardiac hypertrophy and fibrosis.

miR-21 plays a key role in the cardiovascular system regulating cardiac fibrosis. Cardiac stress can determine miR-21 upregulation and activation of ERK/MAPK pathway in cardiac fibroblasts, resulting in fibrosis, cardiac remodeling, and dysfunction [94].

High levels of miR-21 could contribute to fibrosis progression and fibroblast activation in myocardial infarction targeting molecules as PTEN [95,96,97], Smad7 [98], and TGF-β receptor III [99].

Low levels of miR-21 can inhibit fibrotic tissue proliferation in fibroblasts, potentially acting on ERK/MAPK signaling and CADM1 (Cell Adhesion Molecule 1) [95,100].

miR-29, highly expressed in cardiac fibroblasts, is another miRNA involved in fibrosis regulation. It prevents abnormal collagen expression targeting proteins implicated in fibrogenesis such as fibrillins (FBN1), elastin, and multiple collagens (COL1A1, COL1A2, COL3A1) [101,102].

In mouse model, miR-29 inhibits the fibrogenic differentiation of myoblasts into myofibroblasts [103]. Besides, miR-29 addition determined fibrosis decreasing in hypertension [104].

miR-29 could also be involved in the control of cardiac hypertrophy but with contrasting results. A cell model of cardiac hypertrophy and heart failure showed that miR-29 over-expression inhibits cardiomyocyte hypertrophy via inhibiting NFATc4 expression [105]. In contrast, a mouse model of chronic cardiac pressure overload showed that miR-29 deletion inhibits cardiac hypertrophy and fibrosis; miR-29 seems to act, in part, on Wnt signaling pathway inhibiting several cascade factors in cardiomyocytes [106].

miR-133, the third most reported miRNA, is specifically expressed in skeletal muscle, myocardium and cardiac fibroblasts and is involved in myocyte differentiation and cardiac hypertrophy control [107,108,109]. miR-133 can regulate cardiac hypertrophy, both in vivo and in vitro, targeting different genes such as, for example, the β-adrenergic receptor [110], or the hypertrophy-associated mediator NFATc4 [111]. miR-133 overexpression inhibits cardiac hypertrophy while miR-133 suppression induces cardiomyocyte size increasing, without any hypertrophic stimulus [112,113]. miR-133 is also involved in cardiac fibrosis control, regulating the expression of TGF- β1 and TGF- β receptor type II in vivo and in vitro [114]. TGF-β induces CTGF (connective tissue growth factor) expression in connective tissue, controlling collagen production and finally fibrosis. Increased CTGF levels are related to miR-133 low levels while miR-133 overexpression correlates with decreased CTGF levels and also a diminished production of collagens [115].

Despite differences between studies, it is clear that miRNAs are potentially involved in structural inherited cardiomyopathies development and that it is necessary to elucidate the exact role in this context.

Studies conducted on deregulated miRNAs in cardiomyopathies suggest evaluating miRNAs as a potential therapy in treatment of structural cardiac diseases.

Therapy based on the use of miRNAs involves the modulation of miRNAs expression, i.e., using oligonucleotides that mimic miRNA sequence or using miRNAs inhibitors (anti-miR or antago-miR, synthetic antagonists of miRNAs, used for microRNA silencing). miRNA mimics, which are double-stranded oligonucleotides including the mature miRNA sequence and the complementary passenger strand, can be used to restore low miRNA levels [116].

Anti-miRs are instead chemically modified oligonucleotides that bind single-stranded mature miRNAs, preventing target mRNAs binding [116,117].

miRNA mimics and anti-miRs function as therapeutics have been evaluated in animal model of structural cardiomyopathies. For example, the application of miR-302-367 mimics in the mouse model leads to decreased fibrosis, cardiomyocyte proliferation, mass, and improved function after injury [118].

Caré et al. used a miR-133 antagomiR in mice model, resulting in repression of the HCM phenotype [112]. A study conducted on a heart failure HF mice model induced by pressure overload showed inhibition of cardiomyocyte hypertrophy using anti-miR-21 and anti-miR-132 [94]. The reduction of cardiac hypertrophy is observed also in mouse model treated with anti-miR-132 and anti-miR-652 [119,120]. The results obtained in animal model also suggest a possible future evaluation of miRNAs as therapeutics in human cohorts.

## 7. Concluding Remarks

Since their discovery in 1993, miRNAs have been recognized as important members of all biological pathways. During the last few years, miRNA profiling has become an interesting diagnostic research tool, including in the cardiovascular field. miRNAs’ specificity, the possibility of their being co-extracted with DNA, their stability and low susceptibility to degradation are characteristics that make miRNAs ideal biomarkers to be analyzed in clinics.

Our review summarizes the latest news on miRNAs analyzed in human samples of the three most common structural cardiomyopathies. Unfortunately, few miRNAs are in common between the analyzed studies. The diverse results between studies could be related to differences in study design (i.e., sampling mode, extraction methods or quantification) and a possible solution could be the use of standardized protocols.

Research in human samples is in its early stages and further investigations are needed to find common miRNAs to be adopted as validated biomolecules in routine diagnosis, and also to characterize a particular structural cardiomyopathy, to distinguish between cardiomyopathies or phenocopies.

## Figures and Tables

**Table 1 diagnostics-11-01720-t001:** Summary of miRNAs involved in human HCM.

miRNAs in HCM			
Circulating miRNAs			
miRNAs	Quantitative Effect	Notes	Reference
miR-1, miR-16, miR-214	downregulated		[24]
miR-27a, miR-199a-5p, miR-26a, miR-145, mi -133a, miR-143, miR-199a-3p, miR-126-3p, miR-29a, miR-155, miR-30a, miR-21	upregulated	miR-199a-5p, miR-27a correlates with hypertrophy. miR-29a significantly correlates with both cardiac fibrosis and hypertrophy	
miR-373-3p, miR-96-5p	downregulated		[25]
miR-18a-5p, miR-146a-5p, miR-30d-5p, miR-17-5p, miR-200a-3p, miR-19b-3p, miR-21-5p, miR-193-5p, miR-10b-5p, miR-15a-5p, miR-192-5p, miR-296-5p, miR-29a-3p, miR-133a-3p	upregulated	patients with HCM and diffuse myocardial fibrosis	
miR-155	downregulated	miR-155 downregulated in obstructive and non-obstructive HCM	[26]
miR-29a	upregulated	miR-29a upregulated in obstructive HCM with MYH7 mutations	
miR-29c	upregulated	miR-29c high only in aortic stenosis group	
miR-29a	upregulated	HCM patients with fibrosis compared to HCM without fibrosis	[27]
miR-21	upregulated		[28]
**Cardiac miRNAs**			
miR-1, miR-133b, miR-191, miR-208b, miR-218, miR-30b, miR-374, miR-454, miR-495	downregulated	miR-208b downregulated in HCM due to valve disease. miR-495 downregulated in HCM with MYH7 mutations	[29]
miR-590-5p, miR-92a	upregulated		
miR-221	upregulated		[30]
miR-1-3p, miR-23a-3p, miR-23b-3p, miR-24-3p, miR-29b-3p, miR-30d-5p, miR-125a-5p, miR-126-3p, miR-133a-3p, miR-143-3p, miR-145-5p, miR-193b-3p, miR-197-3p, miR-331-3p, miR-342-3p, miR-361-5p, miR-365-3p, miR-455-3p, miR-1975-3p, miR-1978	upregulated	end-stage HCM patients compared to controls	[31]
miR-29 family	-	no differences between patients and controls	[32]
miR-10b, miR10b* miR-184, miR-497, miR-204, miR-222*, miR-34*	downregulated upregulated		[33]
miR-451, miR- 363, miR-150, miR-3141, miR-144, miR-144*, miR-139-5p, miR-139-3p, miR-1246, miR-486-3p miR-21, miR-130b, miR-132	downregulated upregulated		[34]
miR-139-5p	downregulated		[35]
miR-27a, miR-1-3p	downregulated	both miRNAs were downregulated also in DCM cardiac tissue.	[36]
miR-20	upregulated		[37]
miR-19b, miR-155 miR-221, miR-222, miR-433	Downregulated upregulated	in obstructive HCM patients compared to controls	[38]

**Table 2 diagnostics-11-01720-t002:** Summary of miRNAs potentially involved in human ACM.

miRNAs in ACM			
Circulating miRNAs			
miRNAs	Quantitative Effect	Notes	Reference
miR-320a	downregulated	lower in ACM compared to IVT and controls	[51]
miR-144-3p, miR-145-5p, miR-185-5p, and miR-494	upregulated	ACM with ventricular arrhythmia compared to controls	[52]
miR-122-5p, miR-182-5p, and miR-183-5p miR-133a-3p, miR-133b, miR-142-3p	Upregulated downregulated	deregulated in ACM compared to controls and other cardiomyopathies	[53]
**Cardiac miRNAs**			
miR-21-3p, miR-21-5p, miR-34a-5p, miR-212-3p, miR-216a, miR-584-3p, miR-1251, miR-3621, miR-3674, miR-3692-3p, miR-4286, miR-4301	upregulated		[54]
miR-135b, miR-138-5p, miR-193b-3p, miR-302b-3p, miR-302c-3p, miR-338-3p, miR-451a, miR-491-3p, miR-575, miR-3529-5P, miR-4254, miR-4643	downregulated		
miR-29b-3p	upregulated		[55]

**Table 3 diagnostics-11-01720-t003:** Summary of miRNAs potentially involved in human DCM.

miRNAs in DCM			
Circulating miRNAs			
miRNAs	Quantitative Effect	Notes	Reference
miR-142-3p	downregulated	in compensated and in congestive heart failure DCM	[65]
miR-124-5p	upregulated		
miR-548	downregulated	DCM with stable chronic heart failure	[66]
miR-423-5p	upregulated	DCM-related heart failure	[67]
miR-185	upregulated	upregulation linked to a favourable prognosis in DCM patients	[68]
miR-3135b, miR-3908, miR-5571-5p	upregulated		[69]
miR-26, miR-30	similar levels	chronic and new-onset DCM with and without fibrosis	[70]
miR-92b-5p	upregulated	acute heart failure due to DCM	[71]
miR-24-3p, miR-28-5p, miR-100-5p, miR-103-3p, miR-125b5p, miR-214-3p, let-7b-5p, let-7c-5p	upregulated	ischemic and idiopathic DCM	[72]
let-7a-5p, miR-142-3p, miR-145-5p, miR-454-3p	upregulated	DCM with pathogenic LMNA mutations compared to idiopathic DCM or controls	[73]
let-7a-5p, let-7g-5p, miR-16-2-3p, miR-210-3p, miR-215-5p, miR-629-5p	upregulated	in familiar DCM compared to no familiar	[74]
miR-154-5p, miR-182-5p, miR-1249-ep, miR-3191-3p, miR-6769b-3p, miR-6855-5p	upregulated	in DCM patients BAG3+ mutation carriers compared to BAG3 wt	[75]
miR-133a	upregulated	increased expression in LVRR compared to non-LVRR	[76]
miR-130b-3p, miR-150-5p, miR-210-3p	upregulated	distinguish DCM with severe reduced systolic ejection fraction from moderately reduced ejection fraction	[77]
**Cardiac miRNAs**			
miR-214	downregulated		[36]
miR-21	upregulated		
let-7i, miR-126,miR-155	downregulated	let-7i associated with a poor clinical outcome	[78]
miR-21	upregulated		
miR-21	upregulated		[79]
miR-223, miR-650	downregulated	lower in non diabetic DCM than in controls	[80]
miR-216a	upregulated	higher in both DCM groups than in controls	
miR-34b, miR-34c, miR-199b, miR-210	upregulated	higher in diabetic DCM than in controls	
miR-133a	upregulated	inflammatory DCM without fibrosis vs. fibrotic DCM	[81]
miR-378, miR-1, miR-7, miR-29b	downregulated		[82]
miR-342, miR-214, miR-125b, miR-145, miR-181b	upregulated	in end-stage heart failure DCM	
miR-133	downregulated		[83]
miR-21, miR-29	upregulated	in left, right ventricle and apex	
miR-208b	upregulated	high in DCM with severe heart failure and assist device implantation compared to subjects with heart failure due to ischemic heart disease or myocarditis	[84]
miR-26, miR-133a	downregulated		[85]
miR-29	upregulated		

**Table 4 diagnostics-11-01720-t004:** Summary of miRNAs potentially involved in pediatric DCM.

miRNAs in Pediatric DCM			
miRNAs	Quantitative Effect	Notes	Reference
**Circulating miRNAs**			
miR-155, miR-636	upregulated	in subjects with poor prognosis compared to recovered ventricular function	[86]
miR-646, miR-639	downregulated		
miR-454, miR-518f	upregulated		[87]
miR-99b, miR-147, miR-155, miR-194, miR-205, miR-218 miR-302a, miR-544, miR-618, miR-875-3p	downregulated		
let-7f-5p, let-7g-5p, miR-26a-5p, miR-27a-3p, miR-27b-3p, miR-126-3p, miR-142-5p, miR-143-3p	upregulated		[88]
**Cardiac miRNAs**			
miR-29 family	downregulated	in cardiac tissue of heart failure patients without fibrosis	[89]

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
