# Peer review of "MicroRNAs in Hypertrophic, Arrhythmogenic and Dilated Cardiomyopathy"

_diagnostics, 2021, doi:10.3390/diagnostics11091720_

Round 1

Reviewer 1 Report

Dear Authors,

The present review summarizes data on miRNA levels in cardiomyopathies. Wide search for experimental data was performed, however illustrative material and text seem to be confusing and poorly structured.

  1. Authors should totally redesign tables. Discussed studies often include very divergent groups. These data is completely absent in tables, making illustrative material very confusing for reader or even wrong. For example:

- In Table 1, ref 26, miR-29a and miR-29c are marked as “upregulated”. However, the thirst miRNA was shown to be upregulated in obstructive HCM, while the second one was upregulated only in aortic stenosis and not in HCM [26].

- Authors mark miR-34b, miR-34c, miR-199b, miR-210, miR-216a as “upregulated” and miR-223, miR-650 as “downregulated” in DCM [59] in Table 3, although only miR-216a significantly changed its expression in both heart failure groups. Decrease in miR-223 expression was significant only in non-diabetic patients. Other miRNAs were shown to change their expression when diabetic and non-diabetic groups were compared to each other and not with control group.

- In the text authors mention that miR-185  levels are high in patients with favorable prognosis of DCM [69], but mark this miRNA as “upregulated” in Table 3, what looks very confusing. I had to search for the original article to find out that this miRNA was significantly higher in DCM group when compared to controls.

- And so on.

  1. Why are some refs absent in tables, for example, 74, 75, 76, 77?
  2. If authors postulate that their review focuses on “miRNAs expression in human samples” (paragraph 2, line 4), why do they include separate in vitro [37] and in silico [68] studies in the text? If these articles are discussed, other in vitro and in silico studies should be analyzed too.
  3. Since authors suggest using miRNAs as potential biomarkers of cardiomyopathy, I would be very interested in discussing practical possibilities of using cardiac tissue miRNAs, which are widely mentioned in the text, as such biomarkers.
  4. Main text is quite inaccurate. Authors should thoroughly revise it for mistakes, missed words, wrong line breaks, etc.

Author Response

Reviewer #1:

  1. Authors should totally redesign tables. Discussed studies often include very divergent groups. These data is completely absent in tables, making illustrative material very confusing for reader or even wrong. For example:

- In Table 1, ref 26, miR-29a and miR-29c are marked as “upregulated”. However, the thirst miRNA was shown to be upregulated in obstructive HCM, while the second one was upregulated only in aortic stenosis and not in HCM [26].

- Authors mark miR-34b, miR-34c, miR-199b, miR-210, miR-216a as “upregulated” and miR-223, miR-650 as “downregulated” in DCM [59] in Table 3, although only miR-216a significantly changed its expression in both heart failure groups. Decrease in miR-223 expression was significant only in non-diabetic patients. Other miRNAs were shown to change their expression when diabetic and non-diabetic groups were compared to each other and not with control group.

- In the text authors mention that miR-185  levels are high in patients with favourable prognosis of DCM [69], but mark this miRNA as “upregulated” in Table 3, what looks very confusing. I had to search for the original article to find out that this miRNA was significantly higher in DCM group when compared to controls.

- And so on.

We  tried to follow the suggestion of the reviewer and decided to add to the four tables the new column “Notes”, reporting essential information about the references.

- Table 1, ref 26: we specified " miR-155 downregulated in obstructive and non-obstructive HCM. miR-29a upregulated in obstructive HCM with MYH7 mutations ".

- Table 3 ref 80 (Greco S et al.): we rewritten the entire section and specified in Notes of Table3.

- We rewritten ref. 68 Yu M et al.,: " Yu et al. found that miR-185 levels were high in DCM patients compared to controls. Furthermore, high levels of miR-185 were associated with favorable prognosis in DCM patients because this miRNA repress B cells involved in myocardial fibrosis and myocyte injuries [68]" and corrected the appropriate section in the table.

  1. Why are some refs absent in tables, for example, 74, 75, 76, 77?

We thank the reviewer for this comment and added all the missing references to the appropriate table.

  1. If authors postulate that their review focuses on “miRNAs expression in human samples” (paragraph 2, line 4), why do they include separate in vitro[37] and in silico[68] studies in the text? If these articles are discussed, other in vitro and in silico studies should be analyzed too.

We thank the reviewer for this comment and we decided to eliminate the article with comments on in silico and in vitro sections.

  1. Since authors suggest using miRNAs as potential biomarkers of cardiomyopathy, I would be very interested in discussing practical possibilities of using cardiac tissue miRNAs, which are widely mentioned in the text, as such biomarkers.

MiRNAs could certainly represent potential future biomarkers for CM but identify biomarkers for routine practice is still a challenge. Circulating miRNAs could be considered as more useful than cardiac miRNAs because sample collection is easy and less invasive than myocardial biopsy.

  1. Main text is quite inaccurate. Authors should thoroughly revise it for mistakes, missed words, wrong line breaks, etc.

We thank the reviewer for this comment and we tried to do our best correcting all the mistakes regarding the text structure.

Reviewer 2 Report

The paper "microRNAs in structural inherited cardiomyopathies" the role of miRNA in development of cardiomyopathies is described

The paper could be of  interest for the readers of Diagnostics, however some points needs to be addressed

  • HCM

 Underscored that HCM described in the paper is the "sarcomeric form", also add some comments on the role of miRNA in phenocophies.

Diagnosis of HCM in patients with positive family history for HCM is based on a MWT > 13 mm, please add this in the appropriate section .

  • DCM

      Please add some comments on diagnostic criteria of DCM and on the role of ejection fraction in the risk stratification of SCD

Add a section on the role of miRNA in treatment of cardiomyopathies

Add conclusion  

Author Response

Reviewer #2

The paper "microRNAs in structural inherited cardiomyopathies" the role of miRNA in development of cardiomyopathies is described. The paper could be of  interest for the readers of Diagnostics, however some points needs to be addressed.

HCM

  • Underscored that HCM described in the paper is the "sarcomeric form", also add some comments on the role of miRNA in phenocophies.

We followed the suggestion of the reviewer and we added the section 3.3 miRNAs and HCM phenocopies.

  • Diagnosis of HCM in patients with positive family history for HCM is based on a MWT > 13 mm, please add this in the appropriate section.

We followed the suggestion of the reviewer and we added the required text in the appropriate section.

DCM

  • Please add some comments on diagnostic criteria of DCM and on the role of ejection fraction in the risk stratification of SCD.

We thank the reviewer for this remark and we added the required comment in the appropriate section (section 5: miRNAs as potential biomarkers in Dilated Cardiomyopathy).

  • Add a section on the role of miRNA in treatment of cardiomyopathies

We thank the reviewer for this remark and we added this topic at the end of the Discussion.

  • Add conclusion.  

We added “Concluding remarks” as section 7.

Round 2

Reviewer 1 Report

This review is focused on miRNA as potential biomarkers. Although I am fully agree with the fact that circulating miRNAs represent potential future biomarkers, it is not easy to say the same about cardiac miRNAs. Therefore, I still recommend authors to say some words about the role of cardiac miRNAs in clinical use. Maybe they could serve as perspective drug targets or could be used as candidates for search in other, more accessible biomaterials.

Since authors added paragraph 3.3 “miRNA and HCM phenocopies” I recommend reflecting Pubmed search parameters in paragraph 2 “Article selection”.

Were only inherited cardiomyopathies reviewed in the manuscript? If not or authors are not sure, I recommend to eliminate this word from the title and modify abstract and introduction sections.

Author Response

Reviewer 1

This review is focused on miRNA as potential biomarkers. Although I am fully agree with the fact that circulating miRNAs represent potential future biomarkers, it is not easy to say the same about cardiac miRNAs.

  • Therefore, I still recommend authors to say some words about the role of cardiac miRNAs in clinical use. Maybe they could serve as perspective drug targets or could be used as candidates for search in other, more accessible biomaterials.

We tried to answer to the reviewer's request adding this small section in the first part of the paragraph 6. Discussion: "With recent advances in precision medicine, miRNAs that are differentially expressed in cardiac diseases could be considered potential markers. However, circulating miRNAs are more desirable to use as biomarkers in routine diagnosis because they are easily accessible compared to cardiac tissue miRNAs. In fact the complexity in sampling cardiac tissue represent a relevant limit. A possible solution of miRNA tissue analysis could be the evaluation in other biological samples. A promising candidate is represented by urine as suggested by Cheng et al. in a study conducted on rat model of acute myocardial infarction (AMI). Authors found increased levels of heart-released miR-1 and miR-208 in urine samples after AMI [90]. High levels of cardiac-specific miR-1 were also confirmed by Duan et al. in urine of ST elevation acute myocardial infarction (STEMI) patients compared to controls [91]. Furthermore, heart tissue miRNAs could be targets of cardiovascular drugs. Studies conducted on animal model showed that pharmacological treatments modulated miRNAs expression. Zhao et al. analyzed heart tissues of cardiac hypertrophy mouse model and demonstrated that choline administration inhibited pathological cardiac hypertrophy correcting miR-133a downregulation [92]. Tu et al. found instead that atorvastatin inhibited cardiomyocyte hypertrophy in rats downregulating miR-22 and consequently modulating the activity of PTEN [93] ".

  • Since authors added paragraph 3.3 “miRNA and HCM phenocopies” I recommend reflecting Pubmed search parameters in paragraph 2 “Article selection”.

We thank the reviewer for this recommendation and we modified paragraph 2 adding the search term "HCM phenocopies".

  • Were only inherited cardiomyopathies reviewed in the manuscript? If not or authors are not sure, I recommend to eliminate this word from the title and modify abstract and introduction sections.

We thank the reviewer for this suggestion and we decided to change the title as follows: " microRNAs in Hypertrophic, Arrhythmogenic and Dilated Cardiomyopathy".

We also modified the final part of the abstract as follows: " This review summarizes published findings of these molecular biomarkers in the three most common structural cardiomyopathies: human dilated, arrhythmogenic and hypertrophic cardiomyopathy".

Finally, we modified the first part of the Introduction chapter as follows: " Cardiomyopathies are a very heterogeneous group of cardiac diseases, and the inherited forms have a prevalence of...".

Reviewer 2 Report

The authors adressed correctly  the reviewer comments

Author Response

We thank the reviewer for all the previous and precious suggestions